# Relationship between Motor Estimation Error and Physical Function in Patients with Parkinson’s Disease

**DOI:** 10.3390/medicines7080043

**Published:** 2020-07-28

**Authors:** Katsuya Sakai, Tsubasa Kawasaki, Yumi Ikeda, Keita Tominaga, Kohei Kurihara

**Affiliations:** 1Faculty of Healthcare Sciences, Chiba Prefectural University of Health Sciences, Chiba 260-0801, Japan; sakai.katsuya@gmail.com; 2Graduate School of Human Health Sciences, Tokyo Metropolitan University, Tokyo 116-8551, Japan; ikedayum@tmu.ac.jp; 3Institute of Sports Medicine and Science, Tokyo International University, Saitama 350-1198, Japan; 4Third Care Station, Tokyo 191-0001, Japan; kt59311101@yahoo.co.jp; 5Tokyo Rehabilitation Service, Tokyo 133-0057, Japan; koppe923@gmail.com

**Keywords:** motor estimation error, Parkinson’s disease, two-step test, motor planning

## Abstract

**Background**: Motor estimation error is an index of how accurately one’s body movement is recognized. This study determines whether motor estimation error distance is a Parkinson’s disease (PD)- or age-related disability using a two-step task. **Methods**: The participants were 19 PD patients and 58 elderly people with disabilities. A two-step prediction test and an actual two-step test were performed. The motor estimation error distance (prediction of two-step distance minus actual two-step distance) and error rate between the two groups were compared. We conducted a correlation analysis between the motor estimation error and clinical factor (e.g., Hoehn and Yahr stage (H & Y), Unified Parkinson’s Disease Rating Scale (UPDRS)) related to PD. **Results**: The motor estimation error distance was not significantly different between the PD patient group and the elderly group with disabilities. However, significant correlations between motor estimation error and H & Y, and between motor estimation error and UPDRS part II, were observed. The error rate was significantly correlated with the Fall Efficacy Scale. **Conclusions**: The motor estimation error distance is influenced by both aging and PD.

## 1. Introduction

Parkinson’s disease (PD) is a progressive neurodegenerative disease that affects movement. Patients with PD may present with a variety of motor and non-motor symptoms [1,2,3]. The typical motor symptoms are poor balance due to postural reflex disorder, gait disorders, and freezing of gait (FOG). Non-motor symptoms include cognitive disorders, mood disorders (anxiety and depression), sleep disorders, autonomic disorders, hallucinations, and delusions [4].

PD patients experience impaired motor execution due to poor motor planning and motor imagery abilities [5,6]. Cohen et al. [6] reported a decline in motor imagery ability in PD patients. Motor prediction (imagined) and actual duration were measured for the task of walking through a doorway. The results indicated a mismatch between the prediction times and actual walking times when passing through a doorway. Therefore, the study revealed that PD patients have poor motor planning and motor imagery abilities. Similarly, Kamata et al. [7] conducted a functional reach test (FRT) to measure the difference between motor imagery (prediction) and motor execution, showing that overestimation (image distance (i.e., prediction distance) > actual distance) in PD patients was associated with a high fall frequency. The discrepancy between motor imagery and motor execution is referred to as motor estimation error. Motor estimation error indicates the inability to accurately recognize one’s own body movements. Therefore, motor estimation errors in PD patients may be associated with motor disability, fall risk, and gait ability, including FOG.

Elderly people also experience motor estimation error, which correlates with fall risk and physical functions [8,9]. Sakurai et al. [9] investigated whether motor estimation error in a timed up-and-go test differed between community-dwelling older adults with a fear of falling and those without a fear of falling. The results showed that the elderly participants with a fear of falling exhibited larger motor estimation errors. Furthermore, another study investigated the difference in motor estimation error between elderly people who need care or assistance and young people when performing a reaching task [10]. Elderly people who need care or assistance overestimated the task compared to young people. Therefore, motor estimation error was also observed in elderly people who needed care due to physical function impairments and fear of falling.

In a previous study, Kawasaki et al. [11] investigated whether motor estimation error measured using a two-step test differs between patients with PD and healthy elderly people. The results indicated that the motor estimation error was larger in patients with PD than in healthy elderly people. However, in this study, the authors could not clarify whether motor estimation error distance was a PD- or aging-related disability because the control group comprised healthy elderly people. Conversely, in the current study, we investigated whether the motor estimation error distance differs between patients with PD and elderly people with disabilities (the control group) and so were able to determine whether the motor estimation error is a PD- or age-related disability.

Based on these reports, we hypothesize that there could be a difference in motor estimation error between PD patients and elderly people with disabilities when imagining and performing a two-step task. Furthermore, we hypothesize that there is a relationship between motor estimation error and variables related to PD, fear of falling, and anxiety. The purpose of this study was (a) to determine the differences in motor estimation error using a two-step task between PD patients and elderly people with disabilities and (b) to investigate the relationship between motor estimation error and PD-related physical function assessments.

## 2. Materials and Methods

### 2.1. Participants

The participants were 19 patients with PD (Hoehn & Yahr stage (H & Y) of 2–4; mean age, 75.7 ± 8.3, 11 males, 8 females) and 58 elderly people with disabilities (mean age, 77.4 ± 11.9; 25 males, 33 females, Table 1). All participants had no visual deficits, and they could step and maintain position independently without any support. All participants were undergoing rehabilitation at home 1–3 times a week. The field of activities was examined using a Life–Space Assessment (LSA) [12] and was found to be similar between patients with PD and elderly people with disabilities (39.00 ± 28.74 vs. 30.68 ± 21.53 points). The elderly people with disabilities were receiving rehabilitation for disabilities caused by muscle weakness and inactivity in their daily lives. Most elderly people with disabilities were able to walk without support at home.

The purpose of the study was explained to all participants and written informed consent was obtained in compliance with the Declaration of Helsinki. This study was approved by the Institutional Ethics Committee of Tokyo Metropolitan University (approval code: 19026).

### 2.2. Procedure

All clinical assessments were performed during the participants’ home rehabilitation when they were on medication, i.e., periods of little-to-no PD-related disability. The basic characteristics assessed were age, gender, height, weight, and body mass index (BMI: kg/m^2^). Physical functions were measured using a two-step test, two-step prediction test and FRT. For the participants with PD, PD-related assessments were conducted using the Unified Parkinson’s Disease Rating Scale (UPDRS) parts I–IV [13], freezing of gait questionnaire (FOGQ) [14], L-dopa dose (mg/day). In addition, the patients with PD were assessed using the Fall Efficacy Scale (FES) [15] and the Hospital Anxiety and Depression Scale (HADS) [16].

First, participants performed a two-step prediction test, followed by a two-step test (Figure 1). The procedure of the two-step prediction and actual two-step test was explained to the participants by the examiner. For the two-step prediction test, all participants used laser pointers to indicate the predicted two maximum step distance [16]. The prediction distance was then measured using a measure by the examiner. Next, the participants took two maximum steps forward from the line of footing and the actual two-step distance was measured using a measure by the examiner [17,18]. The two-step test has been verified as a reliable test to assess motor function to determine care for elderly people [18].

The motor estimation error (prediction distance (cm) minus actual distance (cm)), error rate (prediction distance (cm)/actual distance (cm) × 100), and two-step value (actual distance/height) were calculated. The two-step value was standardized using height [17].

For the statistical analysis, the Shapiro–Wilk test was used to confirm normal distribution (*p* > 0.05). An unpaired t-test was used to compare whether there was a difference in various clinical variables (age, height, weight, BMI, two-step predicted distance, two-step distance, motor estimation error, error rate, FRT, LSA) between groups. The chi-square test was used to evaluate gender differences.

A Spearman correlation analysis was conducted to determine the relationship between the motor estimation error and other variables (two-step predicted distance, two-step distance, motor estimation error, error rate, FRT, LSA) between the motor estimation error distance and PD-related variables (UPDRS part I–IV, FOGQ, L-dopa dose, FES, HADS). For the H & Y, the correlation coefficient was calculated for each variable using Spearman’s correlation coefficient (two-step predicted distance, two-step distance, motor estimation error, error rate, FRT, LSA, UPPDRS part I–IV, FOGQ, L-dopa dose, FES, HADS). Furthermore, using FES as a confounding factor, a partial correlation analysis was performed to determine the relationship between motor estimation error and PD-related variables.

SPSS software (version 22.0; SPSS Inc., Chicago, IL, USA) was used for all statistical analyses and the statistical significance was set at *p* < 0.05.

## 3. Results

### 3.1. Comparison between Patients with PD and Elderly People with Disabilities

The basic characteristics of patients with PD and elderly people with disabilities are presented in Table 1. There were no significant differences in age, height, weight, BMI, and gender between the two groups (*p* > 0.05).

The results of the physical function assessments are presented in Table 2. The two-step value in the PD patient group was significantly lower than that of the elderly people with disabilities group (t(75) = −2.33, *p* = 0.022). However, the motor estimation error distance and error rate were not significantly different between the two groups (*p* > 0.05). The two-step predicted distance, two-step distance, FRT, and LSA were also not significantly different between the two groups (*p* > 0.05).

### 3.2. Relationship between Motor Estimation Error Distance and Clinical Variables in Patients with PD

The results of the correlation analyses are shown in Table 3. There was a significant positive correlation between motor estimation error distance and H & Y (r = 0.580, *p* = 0.009, Figure 2). There was a significant negative correlation between the error rate and FES (r = −0.470, *p* = 0.042, Figure 3). Furthermore, partial correlation analysis with FES as a control factor revealed a significant positive correlation between motor estimation error distance and UPDRS part II (r = 0.469, *p* = 0.049, Figure 4).

## 4. Discussion

In this study, we investigated the differences in motor estimation error distance between patients with PD and elderly people with disabilities using the two-step test and investigated the relationship between motor estimation error distance and PD-related physical function assessments. The results revealed no significant difference in motor estimation error distance between patients with PD and elderly people with disabilities. The correlation analyses revealed significant correlations between motor estimation error distance and H&Y, and between the error rate and FES. Furthermore, partial correlation analysis with FES as a controlling factor showed that there was a significant positive correlation between motor estimation error distance and UPDRS part II. These findings reveal that the motor estimation error distance is influenced by both aging and PD.

### 4.1. Motor Estimation Error Distance in Patients with PD and Elderly People

Motor estimation error distance was not significantly different between patients with PD and elderly people with disabilities, which was not consistent with our hypothesis. Kawasaki et al. [11] measured estimation error distance using a two-step test in patients with PD and healthy elderly people. The results showed that patients with PD overestimated the distance (image two-step distance > actual two-step distance) more than healthy elderly people. Furthermore, Sakurai et al. [8,9] reported a larger estimation error distance in old elderly people with physical function deficits compared to young elderly people. In the present study, the elderly participants had disabilities caused by muscle weakness and inactivity in their daily lives, which required rehabilitation at home. Therefore, their motor estimation error distance may be larger in comparison to that of healthy elderly people. The reason why no significant difference in the motor estimation error distance was observed between patients with PD and elderly people with disabilities was that the variable was related to both PD- and aging-specific effects.

In addition, Laura et al. investigated the differences in motor imagery ability between patients with PD and healthy people using the finger tapping test [18]. As a result, it was reported that patients with PD had lower motor imagery ability than healthy people. However, other studies report that there is no difference in motor imagery ability between patients with PD and healthy adults [19,20]. These reports consider that some healthy people have low motor imagery ability. Therefore, there is no difference in motor imagery ability between patients with PD and healthy people. In this study, the two-step predicted distance was similar in both groups. Therefore, between patients with PD and elderly people with disabilities, there was no difference in motor imagery ability.

In future research, it will be necessary to longitudinally analyze the motor estimation error distance and clarify the causal relationship.

### 4.2. Relationship between Motor Estimation Error and Clinical Variables in Patients with PD

The motor estimation error distance was related to H & Y. In other words, the overestimation was associated with higher PD severity. Similar to the present study, Kawasaki et al. [11] measured the estimation error distance using a two-step test in PD patients and performed a correlation analysis between the estimation error distance and H&Y. They found no correlation between the estimation error distance and H & Y; however, in this study, we observed a positive correlation. Our study targeted more severe PD patients than the previous studies (H & Y: 3.11 ± 0.67, present study; 2.91 ± 0.70 and 2.50 ± 0.70, previous studies) [11]. H & Y signifies the severity of the motor function deficit in PD patients. Previous studies have demonstrated that the=is large estimation error is related to a decline in motor function [8,9]. In the present study, overestimation was correlated with increasing severity of PD.

Furthermore, there was a significant negative correlation between the error rate and FES. In patients with PD, a large error rate was correlated with a strong fear of falling. The FES consists of 10 questions related to beliefs in their capacity to execute daily life actions without falling [15]. This result is consistent with Sakurai et al. [8,9], who reported that community-dwelling older adults with a fear of falling had a larger estimation error distance compared to those without a fear of falling. In addition, it has been reported that elderly people with a fear of falling have poor balance and walking abilities and a higher incidence of falls [21,22,23]. In patients with PD, it has been reported that fear of falling was related to balance and activities of daily life [24]. These reports demonstrate that a fear of falling reflected physical function.

In addition, motor estimation error distance and UPDRS part II were significantly positively correlated. UPDRS part II expressed the ability to perform activities of daily life [13]. The overestimation by patients with PD was related to lower activity in daily life. Kawasaki et al. [11] also reported that the estimation error distance was correlated with UPDRS part II. Moreover, a correlation between a fear of falling and a lessened ability to perform activities of daily life was reported in patients with PD [24]. These results indicate that a lessened ability to perform activities of daily life occurs when the fear of falling is high. Therefore, as a result of a partial correlation analysis, using FES as a controlling factor, it was speculated that motor estimation error distance and UPDRS might be correlated.

In summary, the motor estimation error distance was not significantly different between patients with PD and elderly people with disabilities. Therefore, it was necessary to consider the influence of aging. However, the motor estimation error distance was correlated with the severity of PD and the UPDRS part II score. Based on these findings, the motor estimation error distance is influenced by both aging and PD.

The main limitations of this study are the small sample size and the lack of a healthy elderly group without disabilities or PD. In the future, we will increase the sample size and recruit participants of H & Y one and five. Moreover, we will investigate whether motor estimation error distance changes longitudinally. The fact that motor estimation error distance is related to the severity of PD and the ability to perform activities of daily life suggests that motor estimation error distance can be a useful tool for easily evaluating the physical function of PD without the need for PD-specific evaluation.

## 5. Conclusions

The results of this study indicate that motor estimation error distance was not significantly different between patients with PD and elderly people with disabilities. However, in patients with PD, the motor estimation error was related to the severity of PD and activities of daily life. Thus, the motor estimation error distance is influenced by both PD and aging.

## Figures and Tables

**Figure 1 medicines-07-00043-f001:**
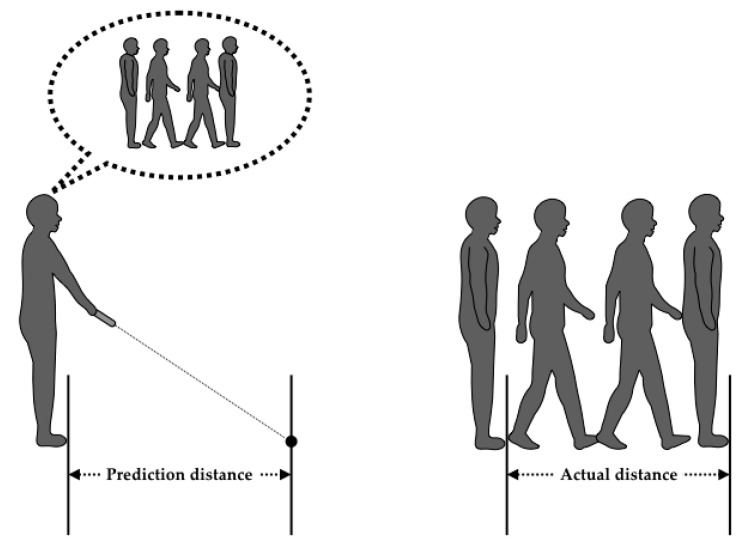
Illustration of the two-step prediction test and actual two-step test. For the two-step prediction test, the participants used laser pointers to predict the maximum two-step distance. After the two-step prediction test, the actual two-step distance was measured.

**Figure 2 medicines-07-00043-f002:**
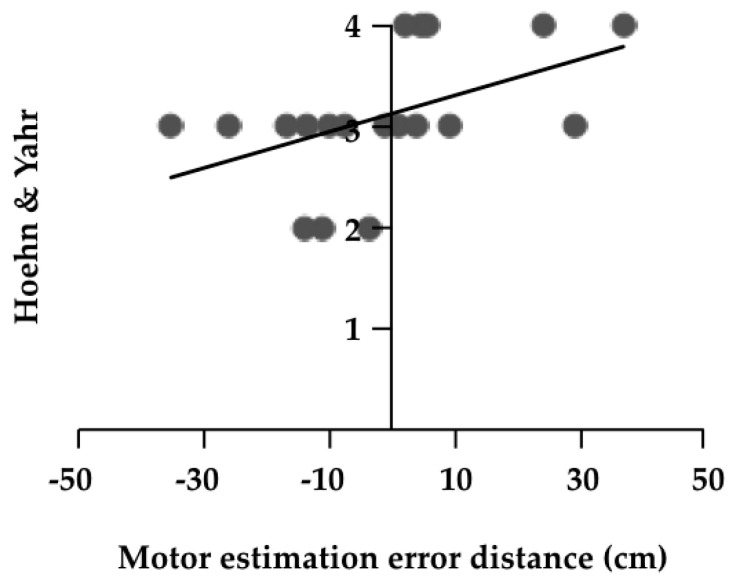
Correlation coefficient of the motor estimation error distance and Hoehn & Yahr stage.

**Figure 3 medicines-07-00043-f003:**
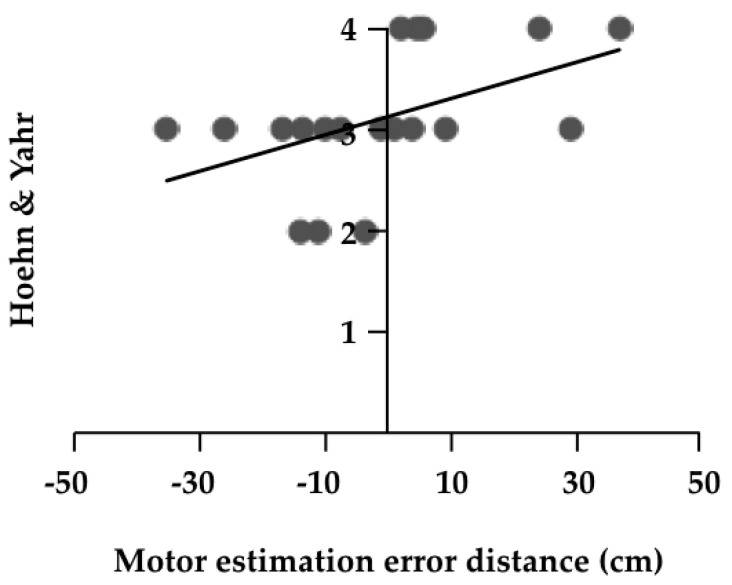
Correlation coefficient of the motor estimation error distance and Unified Parkinson’s Disease Rating Scale part II. Unified Parkinson’s Disease Rating Scale (UPDRS).

**Figure 4 medicines-07-00043-f004:**
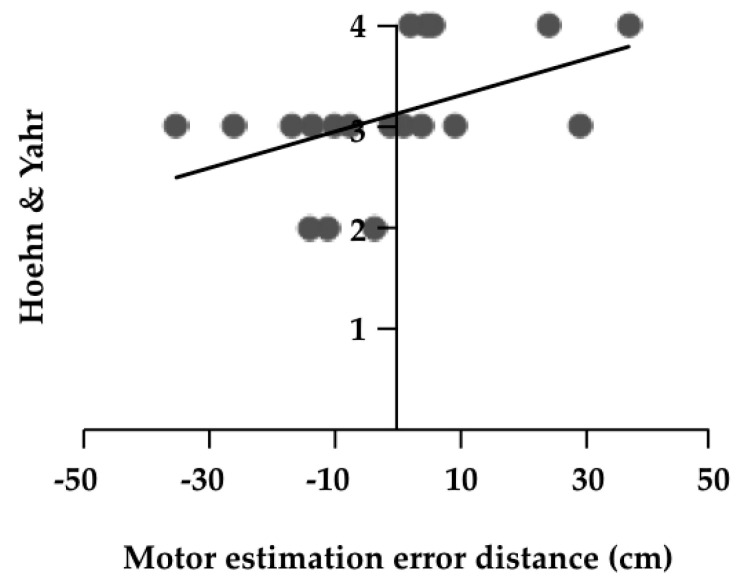
Correlation coefficient of the error rate and fall efficacy scale. Fall Efficacy Scale (FES).

**Table 1 medicines-07-00043-t001:** Basic characteristics of study participants.

Basic Characteristics	Patients with PD (n = 19)	Elderly People with Disabilities (n = 58)	*p* Value
Age (years)	75.68 ± 8.26	77.40 ± 11.89	0.562
Males/Females	11/7	25/33	0.196
Height (cm)	155.30 ± 8.74	157.11 ± 10.46	0.499
Weight (kg)	51.29 ± 9.09	55.49 ± 11.78	0.160
BMI (kg/m^2^)	21.18 ± 2.75	22.40 ± 4.00	0.219
H & Y	3 (2–4)		-

Parkinson’s disease (PD), body mass index (BMI), Hoehn & Yahr stage (H & Y).

**Table 2 medicines-07-00043-t002:** Physical function and Parkinson’s disease-related assessments.

Assessments	Patients with PD (n = 19)	Elderly People with Disabilities (n = 58)	*p* Value
Two-step distance (cm)	96.74 ± 34.73	100.26 ± 40.42	0.734
Two-step value	0.62 ± 0.21	1.09 ± 0.86	0.022 *
Two-step predicted distance (cm)	95.68 ± 38.11	102.18 ± 44.30	0.575
Motor estimation error distance (cm)	−1.06 ± 18.09	1.91 ± 24.34	0.627
Error rate (%)	15.76 ± 12.63	18.31 ± 18.74	0.582
FRT (cm)	18.74 ± 6.55	18.01 ± 6.61	0.679
LSA (points)	39.00 ± 28.74	30.68 ± 21.53	0.184
UPDRS part I (points)	1.36 ± 1.77	-	-
UPDRS part II (points)	11.05 ± 8.06	-	-
UPDRS part III (points)	13.63 ± 10.13	-	-
UPDRS part IV (points)	3.31 ± 3.11	-	-
UPDRS total (points)	29.36 ± 19.57	-	-
FOGQ (points)	10.05 ± 5.99	-	-
L-dopa dose (mg/day)	435.76 ± 282.54	-	-
FES (points)	26.00 ± 8.32	-	-
HADS anxiety (points)	2.83 ± 2.73	-	-
HADS depression (points)	5.39 ± 4.79	-	-

* *p* < 0.05. Parkinson’s disease (PD), functional reach test (FRT), Life–Space Assessment (LSA), Unified Parkinson’s Disease Rating Scale (UPDRS), freezing of gait questionnaire (FOGQ), Fall Efficacy Scale (FES), Hospital Anxiety and Depression Scale (HADS).

**Table 3 medicines-07-00043-t003:** Correlation coefficients for patients with Parkinson’s disease.

Assessments	Assessments	Correlation Coefficient	*p* Value
Two-step distance	Two-step value	0.983	0.001
	Two-step predicted distance	0.881	0.001
	FRT	0.469	0.043
	LSA	0.628	0.004
Two-step value	Two-step predicted distance	0.902	0.001
	LSA	0.562	0.012
Two-step predicted distance	LSA	0.692	0.001
	H & Y	0.469	0.043
Motor estimation error distance	H & Y	0.580	0.009
	UPDRS part II	0.469	0.049
Error rate	FES	−0.470	0.042
FRT	FES	0.522	0.022
LSA	FES	0.585	0.009
UPDRS part II	UPDRS part III	0.851	0.001
	UPDRS total	0.926	0.001
	FOGQ	0.555	0.014
	FES	−0.482	0.037
UPDRS part III	UPDRS part IV	0.499	0.030
	UPDRS total	0.962	0.001
	FES	−0.508	0.026
UPDRS part IV	UPDRS total	0.594	0.007
	FOGQ	0.544	0.016
	L-dopa dose	0.727	0.001
	HADS anxiety	0.798	0.001
UPDRS total	FOGQ	0.583	0.009
	L-dopa dose	0.504	0.028
	FES	−0.517	0.023
	HADS anxiety	0.507	0.027
FOGQ	L-dopa dose	0.467	0.044
L-dopa dose	HADS anxiety	0.686	0.001

Hoehn and Yahr stage (H & Y), functional reach test (FRT), Life–Space Assessment (LSA), Unified Parkinson’s Disease Rating Scale (UPDRS), freezing of gait questionnaire (FOGQ), Fall Efficacy Scale (FES), Hospital Anxiety and Depression Scale (HADS).

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
