# Peer review of "Relationship between Motor Estimation Error and Physical Function in Patients with Parkinson’s Disease"

_medicines, 2020, doi:10.3390/medicines7080043_

Round 1
Reviewer 1 Report
It is necessary to justify the relevance of the objectives of this research. You have not done it.
Further explanation of the sample is needed, mainly in non-parkinson patients. and it is also necessary to explain whether your comparison is appropriate
the "n" of the samples is very different, it would be appropriate to readjust the statistical analysis.
Author Response
Response to Reviewer
We wish to express our appreciation to the reviewers for their insightful comments, which have helped us significantly improve the paper. We have provided point-by-point responses to their comments and indicated the changes made in the revised manuscript.
Comments 1:
It is necessary to justify the relevance of the objectives of this research. You have not done it.
Response 1:
We agree with your comments. We described the objectives of this research.
In a previous study, Kawasaki et al .investigated whether the motor estimation error distance measured using the two-step test differed between patients with Parkinson’s disease (PD) and healthy elderly people [1]. In this comparison, it was not possible to determine whether the motor estimation error distance is a PD-specific disability or a disability associated with aging. Therefore, in the current study, we aimed to clarify whether the motor estimation error distance is a PD- or age-related disability.
Reference
[1] Kawasaki, T.; Mikami, K.; Kamo, T.; Aoki, R.; Ishiguro, R.; Nakamura, H.; Tozawa, R.; Asada, N.; Hiiragi Y.; Yamada, Y.; Hirano, M.; Katsuki, K. Motor planning error in Parkinson’s disease and its clinical correlates. PLoS One. 2018, 13, e0202228. doi: 10.1371/journal.pone.0202228.
Page 2, lines 66–75
In a previous study, Kawasaki et al. [11] investigated whether the motor estimation error measured using the two-step test differs between patients with PD and healthy elderly people. The results indicated that the motor estimation error was larger in patients with PD than in healthy elderly people. However, in this study could not clarify whether motor estimation error distance was a PD- or aging-related disability because the control group comprised healthy elderly people. Conversely, in the current study, we investigated whether the motor estimation error distance differs between patients with PD and elderly people with disabilities (the control group) and so were able to determine whether the motor estimation error is a PD- or age-related disability.
Page 9, lines 264–269
In summary, the motor estimation error distance was not significantly different between patients with PD and elderly people with disabilities. Therefore, it was necessary to consider the influence of aging. However, the motor estimation error distance was correlated with the severity of PD and the UPDRS part II score. Based on these findings, the motor estimation error distance is influenced by both aging and PD.
Page 10, lines 278–282
The results of this study indicate that motor estimation error distance was not significantly different between patients with PD and elderly people with disabilities. However, in patients with PD, the motor estimation error was related to the severity of PD and activities of daily living. Thus, the motor estimation error distance is influenced by both PD and aging.
Comments 2;
Further explanation of the sample is needed, mainly in non-parkinson patients. And it is also necessary to explain whether your comparison is appropriate.
Response 2;
We added a description of the functional level of the elderly people with disabilities. In addition, we stated that patients with PD and elderly people with disabilities have similar life–space (field of activities).
The appropriateness of the comparison is described in Response 1.
Page 3, lines 86–96
The participants were 19 patients with PD [Hoehn & Yahr stage (H & Y) of 2–4; mean age, 75.7 ± 8.3, 11 males, 8 females] and 58 elderly people with disabilities (mean age, 77.4 ± 11.9; 25 males, 33 females, Table 1). All participants had no visual deficits, and they could step and maintain position independently without any support. All participants were undergoing rehabilitation at home 1–3 times a week. The field of activities was examined using Life–Space Assessment (LSA) [12] and was found to be similar between patients with PD andelderly people with disabilities (39.00 ± 28.74 vs 30.68 ± 21.53 points). The elderly people with disabilities were receiving rehabilitation caused by muscle weakness and inactivity of daily living. Most elderly people with disabilities were able to walk without support at home.
Comments3;
The "n" of the samples is very different. It would be appropriate to readjust the statistical analysis.
Response 3;
We appreciate this comment. However, some previous studies of the motor estimation error distance using similar statistical analyses as our study were performed using different samples.
Sakurai et al. compared young (n = 71), middle-age (n = 343), and elderly adults (n = 151) [1]. The statistical analysis in this study was performed as described in previous studies. In addition, an unpaired t-test was performed to confirm normality using the Shapiro–Wilk test, and the results of this test have been added.
Page 4, line 130-131
For the statistical analysis, the Shapiro–Wilk test was used to confirm normal distribution (p > 0.05).
Reference
[1] Sakurai, R.; Fujiwara, Y.; Ishihara, M.; Higuchi, T.; Uchida, H.; Imanaka, K. Age-related self-overestimation of step-over ability in healthy older adults and its relationship to fall risk. BMC Geriatr.2013, 13, 44. doi: 10.1186/1471-2318-13-44.
Reviewer 2 Report
Thanks to the authors for providing an aspect of interesting measurements in PD. I think the paper was written clearly encouraging the reader to follow. However the below notes are proposed to authors for improvement of the paper:
1- The results are the core of the paper. The correlations were not high enough to make the paper as an extensively interesting one.
2- In the results section, the table 1 to be presented in participation section together with the p_values.
3- Table 2, from Hoen & Yahr to end belongs to the participants section.
4- Table 3, the title to be revised, and the content are too many. Authors may report only the significant correlations.
5- If the correlations are reported I'm not sure the Fig.2 is needed at all.
6- In the conclusion the authors mentioned that the hypothesis is not met. They may express how the results can be used in practice. Specially if the Correlations are not high.
7- In Introduction, authors may express about how this study differs from the other similar papers reviewed.
8- In PD, the cognition might be disrupted. How are the correlations to UPDRS part-I the 1.1 cognitive impairment? Then if it agrees with those items why is this study found useful if cognitive impairment can be measured by UPDRS?
9- In method, it was not described how the UPDRS scores measured? were they the mean all items from each section? median?
10- Measuring two step value (actual distance/height) is not described adequately. Why height is considered?
11- In methods, what was the software used for measuring the physical functions?
12- Authors to express about why and how the results can be useful? Have they considered employing machine learning methods? are there any potential devices to be developed using these results?
13- The novelty of the paper to be highlighted.
Author Response
Response to Reviewer
We wish to express our appreciation to the reviewers for their insightful comments, which have helped us significantly improve the paper. We have provided point-by-point responses to their comments and indicated the changes made in the revised manuscript.
Thanks to the authors for providing an aspect of interesting measurements in PD. I think the paper was written clearly encouraging the reader to follow. However the below notes are proposed to authors for improvement of the paper:
Comments 1;
1- The results are the core of the paper. The correlations were not high enough to make the paper as an extensively interesting one.
Response 1;
We thank you for this comment. The correlation of the main results in this study was moderate or high (between motor estimation error distance and the Hoehn & Yahr stage: r = 0.580, between the error rate and FES: r = −0.470, between the motor estimation error distance and UPDRS part II score: r = 0.469). However, this study did not emphasize the high degree of correlation. An important point of this study was the observation of a correlation between the motor estimation error distance and disease severity (Hoehn & Yahr). Kawasaki et al. reported no significant correlation between the motor estimation error distance and disease severity [1]. However, in this study, the motor estimation error distance was correlated with disease severity. Our study targeted a higher number of patients with severe Parkinson’s disease (PD) than previous studies (Hoehn & Yahr: 3.11 ± 0.67, present study; 2.91 ± 0.70 and 2.50 ± 0.70,Kawasaki et al.). Therefore, the analyses revealed significant correlations between the motor estimation error distance and disease severity. Moreover, it will be important to conduct longitudinal analysis in future research.
Reference
[1] Kawasaki, T.; Mikami, K.; Kamo, T.; Aoki, R.; Ishiguro, R.; Nakamura, H.; Tozawa, R.; Asada, N.; Hiiragi Y.; Yamada, Y.; Hirano, M.; Katsuki, K. Motor planning error in Parkinson’s disease and its clinical correlates. PLoS One. 2018, 13, e0202228. doi: 10.1371/journal.pone.0202228.
Comments 2;
2- In the results section, the table 1 to be presented in participation section together with the p_values.
Response 2;
Table 1 was moved to the participation section.
Comments 3;
3- Table 2, from Hoen & Yahr to end belongs to the participants section.
Response 3;
The Hoehn & Yahr stage was added to the participants section and Table 1.
Comments 4;
4- Table 3, the title to be revised, and the content are too many. Authors may report only the significant correlations.
Response 4;
We only listed variables with significant correlations in Table 3.
Comments 5;
5- If the correlations are reported I'm not sure the Fig.2 is needed at all.
Response 5;
We agree with your comments. However, we believe that the figure improves the understanding of our findings. Therefore, we would like to retain this figure in the manuscript.
Comments 6;
6- In the conclusion the authors mentioned that the hypothesis is not met.
They may express how the results can be used in practice. Specially if the Correlations are not high.
Response 6;
We agree with your comments. In this study, the motor estimation error distance and error rate were not significantly different between the two groups. This result indicated that the motor estimation error distance was related to both PD- and aging-specific disability. However, the correlation analyses revealed significant correlations between the motor estimation error distance and disease severity. Therefore, it will be important to conduct longitudinal analysis in future research.
Comments 7;
7- In Introduction, authors may express about how this study differs from the other similar papers reviewed.
Response 7;
This study examined whether the motor estimation error distance is a PD- or age-related disability. In this effort, we investigated whether the motor estimation error distance differs between patients with PD and elderly people with disabilities. In a previous study, Kawasaki et al. [1] investigated whether the motor estimation error distance measured using the two-step test differed between patients with PD and healthy elderly people. The results indicated that the motor estimation error distance was larger in patients with PD than in healthy elderly people. Their study did not clarify whether the motor estimation error distance was a PD- or aging-related disability.
Reference
[1] Kawasaki, T.; Mikami, K.; Kamo, T.; Aoki, R.; Ishiguro, R.; Nakamura, H.; Tozawa, R.; Asada, N.; Hiiragi Y.; Yamada, Y.; Hirano, M.; Katsuki, K. Motor planning error in Parkinson’s disease and its clinical correlates. PLoS One. 2018, 13, e0202228. doi: 10.1371/journal.pone.0202228.
Page 2, lines 66–75
In a previous study, Kawasaki et al. [11] investigated whether the motor estimation error measured using the two-step test differs between patients with PD and healthy elderly people. The results indicated that the motor estimation error was larger in patients with PD than in healthy elderly people. However, in this study could not clarify whether motor estimation error distance was a PD- or aging-related disability because the control group comprised healthy elderly people. Conversely, in the current study, we investigated whether the motor estimation error distance differs between patients with PD and elderly people with disabilities (the control group) and so were able to determine whether the motor estimation error is a PD- or age-related disability.
Comments 8;
8- In PD, the cognition might be disrupted. How are the correlations to UPDRS part-I the 1.1 cognitive impairment? Then if it agrees with those items why is this study found useful if cognitive impairment can be measured by UPDRS?
Response 8;
No correlation was found between the UPDRS part II score and the motor estimation error distance. Moreover, the UPDRS part II score of patients with PD was 1.36 ± 1.77 points. Therefore, cognitive impairment did not influence the motor estimation distance.
Comments 9;
9- In method, it was not described how the UPDRS scores measured? were they the mean all items from each section? median?
Response 9;
We have rewritten the text on Page 4, line 104-106:
All clinical assessments were performed during the participants’ home rehabilitation when they were on medication, i.e., periods of little-to-no PD-related disability.
The UPDRS score is often used as the average value. We used the average value according to a previous study [1].
Reference
[1] Kawasaki, T.; Mikami, K.; Kamo, T.; Aoki, R.; Ishiguro, R.; Nakamura, H.; Tozawa, R.; Asada, N.; Hiiragi Y.; Yamada, Y.; Hirano, M.; Katsuki, K. Motor planning error in Parkinson’s disease and its clinical correlates. PLoS One. 2018, 13, e0202228. doi: 10.1371/journal.pone.0202228.
Comments 10;
10- Measuring two step value (actual distance/height) is not described adequately. Why height is considered?
Response 10;
The step distance depends on the height. In a previous study [1], height was divided for the purpose of standardization.
Page 4, lines 128–129
Two-step value was standardized using height [17].
Comments 11;
11- In methods, what was the software used for measuring the physical functions?
Response 11;
No software was used to measure physical function; it was measured using a measure by the examiner during the participants’ home rehabilitation.
Page 4, lines 117-120
The prediction distance was then measured using a measure by the examiner. Next, the participants took two maximum steps forward from the line of footing and the actual two-step distance was measured using a measure by the examiner [17,18].
Comments 12;
12- Authors to express about why and how the results can be useful? Have they considered employing machine learning methods? are there any potential devices to be developed using these results?
Response 12;
Machine learning will be considered in a future study. However, it has not been clarified whether the motor estimation error is a symptom specific to PD. In the future, we would like to clarify the relationship between the motor estimation error distance and PD based on the results of this study.
If a longitudinal analysis of the motor estimation error distance is performed and the relationship between the motor estimation error distance and physical function is clarified, it may be possible to develop devices that use the estimation error as a predictive factor.
Comments 13;
13- The novelty of the paper to be highlighted.
Response 13;
The novelty of the paper was that the motor estimation error distance was determined to be related to both PD and aging.
Page 9-10, lines 264-269
In summary, the motor estimation error distance was not significantly different between patients with PD and elderly people with disabilities. Therefore, it was necessary to consider the influence of aging. However, the motor estimation error distance was correlated with the severity of PD and the UPDRS part II score. Based on these findings, the motor estimation error distance is influenced by both aging and PD.
Round 2
Reviewer 1 Report
I think it is necessary to be able to justify the work they have done without the only argument being to rely on other studies. Other bibliographic references that present the opposite can be found.
Author Response
Reviewer 1
Response to Reviewer
We wish to express our appreciation to the reviewers for their insightful comments, which have helped us significantly improve the paper. We have provided point-by-point responses to their comments and indicated the changes made in the revised manuscript.
Comment:
I think it is necessary to be able to justify the work they have done without the only argument being to rely on other studies. Other bibliographic references that present the opposite can be found.
Response:
We appreciate this comment. In this study, we investigated whether the motor estimation error distance differs between patients with Parkinson’s disease (PD) and elderly people with disabilities (the control group) and were able to determine whether the motor estimation error is a PD- or age-related disability. This study determined that motor estimation error distance was associated with both aging and PD. This study gives the important result that it is associated with the motor estimation error distance and the specific evaluation of PD. Kawasaki et al., reported not significant correlation between motor estimation error distance and severity [1]. However, in this study, motor estimation error distance was correlated the severity. The result is the novelty of this study.
In addition, Laura et al. investigated the differences in motor imagery ability between patients with PD and healthy people using the finger tapping test [2]. As a result, it was reported that patients with PD had lower motor imagery ability than healthy people. However, other studies report that there is no difference in motor imagery ability between patients with PD and healthy adults [3, 4]. These reports consider that some healthy people have low motor imagery. Therefore, there is no difference in motor imagery ability between patients with PD and healthy people. In this study, the prediction two-step distance was similar in both groups. Therefore, patients with PD and elderly people with disabilitiesindicated that there was no difference in motor imagery ability.
The limitations of this study are the small sample size and the lack of a healthy elderly group without disabilities or PD. In the next study, we will improve this limitation and further explore the relationship between motor estimation error distance and PD.
Reference
[1] Kawasaki, T.; Mikami, K.; Kamo, T.; Aoki, R.; Ishiguro, R.; Nakamura, H.; Tozawa, R.; Asada, N.; Hiiragi Y.; Yamada, Y.; Hirano, M.; Katsuki, K. Motor planning error in Parkinson's disease and its clinical correlates. PLoS One. 2018, 13, e0202228. doi: 10.1371/journal.pone.0202228.
[2] Avanzino, L.; Pelosin, E.; Martino, D.; Abbruzzese, G. Motor timing deficits in sequential movements in Parkinson disease are related to action planning: a motor imagery study. PLoS ONE. 2013, 8, e75454. doi: 10.1371/journal.pone.0075454.
[3] Pickett. K. A.; Peterson, D.S.; Earhart, G.M. Motor imagery of gait tasks in individuals with Parkinson disease. J Parkinsons Dis. 2012, 2, 19-22. doi: 10.3233/JPD-2012-11045.
[4] Peterson, D.S.; Pickett, K.A.; Earhart, G. M. Effects of levodopa on vividness of motor imagery in Parkinson disease. J Parkinsons Dis. 2012, 2, 127-133. doi: 10.3233/JPD-2012-12077.
Page 9, lines 226 – 235.
In addition, Laura et al. investigated the differences in motor imagery ability between patients with PD and healthy people using the finger tapping test [18]. As a result, it was reported that patients with PD had lower motor imagery ability than healthy people. However, other studies report that there is no difference in motor imagery ability between patients with PD and healthy adults [19, 20]. These reports consider that some healthy people have low motor imagery. Therefore, there is no difference in motor imagery ability between patients with PD and healthy people. In this study, the prediction two-step distance was similar in both groups. Therefore, patients with PD and elderly people with disabilitiesindicated that there was no difference in motor imagery.
Reviewer 2 Report
Thanks to the authors for taking the efforts for updating the paper. Most of the comments are answered. However, the future work of this paper can be enhanced towards the better achievements in treatment of PD and gaining higher correlations.
Author Response
Reviewer 2
Response to Reviewer
We wish to express our appreciation to the reviewers for their insightful comments, which have helped us significantly improve the paper. We have provided point-by-point responses to their comments and indicated the changes made in the revised manuscript.
Comment:
Thanks to the authors for taking the efforts for updating the paper. Most of the comments are answered. However, the future work of this paper can be enhanced towards the better achievements in treatment of PD and gaining higher correlations.
Response:
We appreciate this comment. In this study, patients with Parkinson’s disease (PD) were Hoehn & Yahr stage (H & Y) 2 – 4. Therefore, a moderate correlation was found between motor estimation error distance and severity of PD. In the future study, it is assumed that the correlation coefficient will become higher by increasing the number of participants and by investigating patients with PD with wide range of severity (H & Y 1 and 5 in addition to H & Y 2 − 4). We conclude that this result showed the important result that motor estimation error distance could be an evaluation tool specific to patients with PD in clinical settings.
Page 10, lines 281−283.
We will increase the sample size and recruite participants of H & Y 1 and 5. Moreover, we investigate whether motor estimation error distance changes longitudinally.